# Chiral phosphoric acid-catalyzed asymmetric epoxidation of alkenyl aza-heteroarenes using hydrogen peroxide

Hao-Chen Wen[1,4], Wei Chen[1,4], Meng Li[1,4], Chen Ma[2], Jian-Fei Wang[1], Aiping Fu[1], Shi-Qi Xu[1], Yi-Feng Zhou[3], Shao-Fei Ni [2] ✉ & Bin Mao [1] ✉

The synthesis of chiral α-azaheteroaryl oxiranes via enantioselective catalysis is a formidable challenge due to the required complex stereoselectivity and diverse N-heterocyclic structures. These compounds play a crucial role in developing bioactive molecules, where precise chirality significantly influences biological activity. Here we show that using chiral phosphoric acid as a catalyst, our method efficiently addresses these challenges. This technique not only achieves high enantio- and diastereoselectivity but also demonstrates superior chemo- and stereocontrol during the epoxidation of alkenyl aza-heteroarenes. Our approach leverages a synergistic blend of electrostatic and hydrogen-bonding interactions, enabling the effective activation of both substrates and hydrogen peroxide. The resulting chiral oxiranes exhibit enhanced diversity and functionality, aiding the construction of complex chiral azaaryl compounds with contiguous stereocenters. Kinetic and density functional theory studies elucidate the mechanism, highlighting chiral phosphoric acid's pivotal role in this intricate enantioselective process.

Over the last three decades, the field of asymmetric epoxidation has seen substantial advancements[1–9], profoundly impacting the synthesis of optically active epoxides essential in diverse chemical processes. The period has witnessed major breakthroughs in chiral metal catalysis[10–13] and organocatalysis[14,15], including the titanium-catalyzed Sharpless epoxidation[10], Jacobsen-Katsuki epoxidation[11], and Shi's fructose-based epoxidation[14], targeting electron-neutral or electron-rich olefins with electrophilic oxidants effectively (Fig. 1a). In parallel, developments in asymmetric nucleophilic epoxidation, exemplified by the Weitz-Scheffer epoxidation[16], have leveraged chiral ligand-enhanced metal peroxides[17,18], polypeptides[19], cinchona-based alkaloids[20] as catalyst components, significantly expanding substrate versatility and achieving high enantioselectivity. Recent strides in covalent aminocatalysis, led by Jørgensen[21], MacMillan[22], Córdova[23] and notably List[24–27], have further diversified the epoxidation scope,

encompassing various enones and enals through the activation of iminium ion intermediates (Fig. 1b). Despite the considerable progress, the enantioselective epoxidation of alkenyl N-heteroarenes, which is pivotal for the creation of chiral α-azaaryl oxiranes, continues to present a formidable challenge[28,29]. The intrinsic limitations of azaarenes, including their low electron-withdrawing capacity and propensity towards N-oxide formation, significantly impede conventional epoxidation techniques. Additionally, while non-covalent organocatalysis has shown preliminary success in epoxidizing electron-deficient olefins[30–35], significant challenges related to efficiency and selectivity remain unresolved, particularly in the context of complex heterocyclic systems. These ongoing challenges underscore an urgent need for innovative epoxidation strategies capable of surmounting these barriers and bridging a notable gap in existing synthetic repertoire.

[1]Collaborative Innovation Center of Yangtze River Delta Region Green Pharmaceuticals, Zhejiang University of Technology, Hangzhou, P.R. China. [2]Department of Chemistry and Key Laboratory for Preparation and Application of Ordered Structural Materials of Guangdong Province, Shantou University, Shantou, China. [3]College of Life Science, China Jiliang University, Hangzhou, P.R. China. [4]These authors contributed equally: Hao-Chen Wen, Wei Chen, Meng Li. ✉e-mail: sfni@stu.edu.cn; maob@zjut.edu.cn

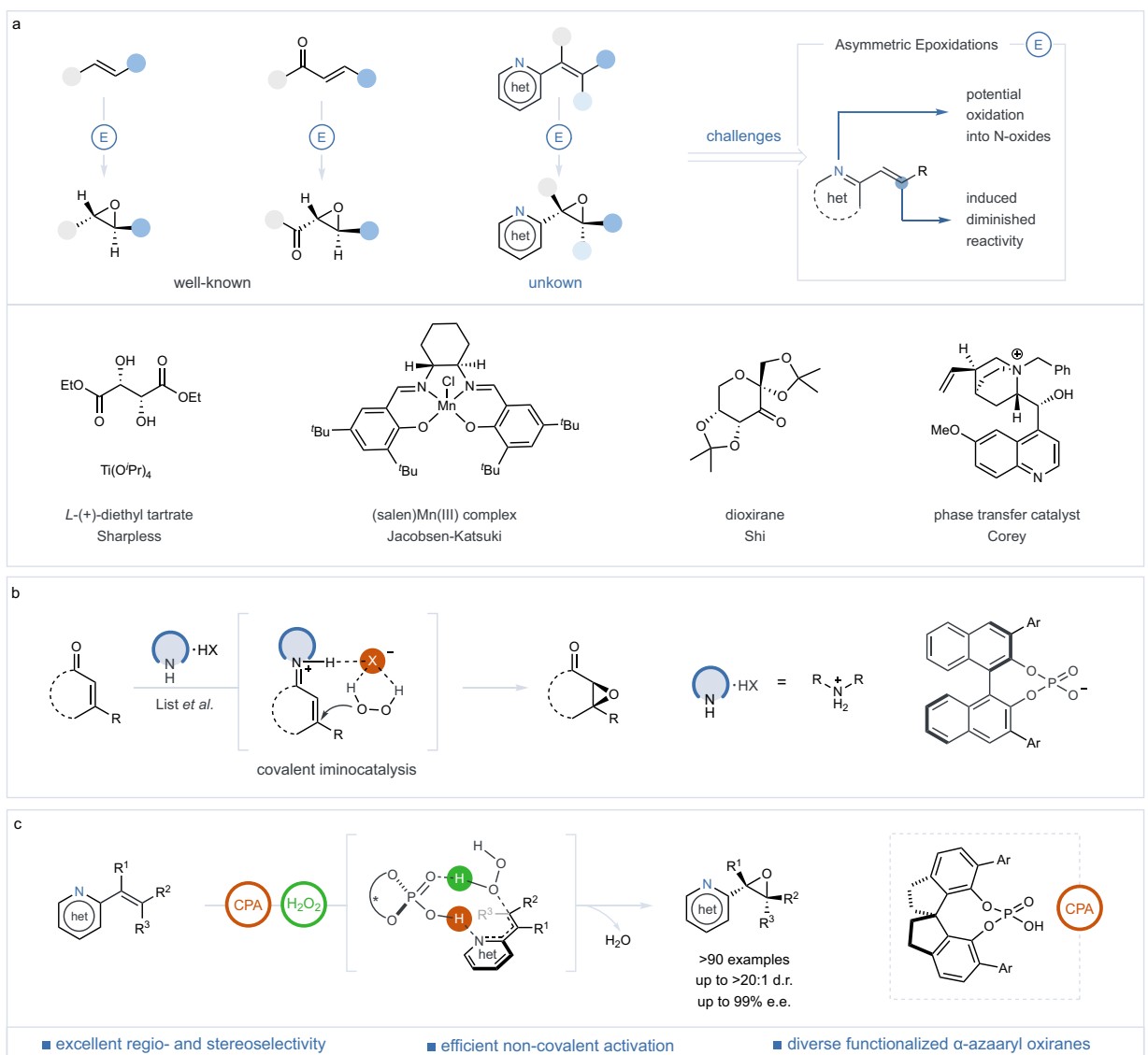

**Fig. 1 | Organocatalytic asymmetric epoxidations of alkenyl aza-heteroarenes.**
**a** Survey of prior achievements and unaddressed challenges in the catalytic asymmetric epoxidation. **b** Mechanistic insights into asymmetric epoxidation via covalent aminocatalysis, with a focus on the formation of chiral ion-pair intermediates. **c** Our approach involves enantioselective epoxidation of alkenyl *N*-heteroarenes catalyzed by chiral phosphoric acid using hydrogen peroxide as oxidant. CPA, chiral phosphoric acid.

Chiral azaarenes, especially those with imine functionalities, hold critical importance in pharmaceutical research due to their significant impact on drug efficacy and selectivity[36]. The pursuit of enantioenriched azaarene derivatives has led to heightened research into catalytic asymmetric reactions[37–39], focusing on methods like conjugate addition to prochiral azaarene-derived substrates[40–47] and enantioselective addition to 2-alkylazaarenes[48–51]. Despite the strides made with photoredox catalysis exploiting azaarenes' unique attributes for radical-based synthesis[52–58], the seamless integration of vicinal stereogenic centers into nitrogen-containing heteroarenes has remained a significant challenge[59,60]. Traditional techniques often fall short in efficiency and specificity when modifying complex structures such as α-azaaryl oxiranes and α-cyanomethylazaarenes, which are essential both as synthetic intermediates and as bases for pharmaceuticals[61–64]. Addressing this gap, the advent of direct, catalytic asymmetric synthesis heralds a considerable advancement, introducing a targeted and precise method to instill chirality and functionality directly into azaaryls. This streamlined approach promises to revolutionize synthetic processes and significantly impact the pharmaceutical industry.

In this study, we address the enduring challenges in the enantioselective epoxidation of alkenyl *N*-heteroarenes through the deployment of a chiral Brønsted acid catalysis method (Fig. 1c). Drawing on the principles of asymmetric counteranion-directed catalysis (ACDC)[65,66], our approach employs chiral phosphoric acid for the simultaneous activation of aza-heteroarene substrates and hydrogen peroxide. This dual activation approach significantly enhances the electrophilic nature of the alkenyl group through the protonation of adjacent *N*-heteroarenes and increases the nucleophilicity of hydrogen peroxide. Consequently, it transcends the conventional limitations associated with azaarenes, enabling selective epoxidation across a diverse array of alkenyl *N*-heteroarenes, including those devoid of electron-withdrawing groups. The method promotes efficient organocatalytic nucleophilic epoxidation, delivering unparalleled control over chemo- and stereoselectivity via attractive non-covalent interactions. This innovative technique not only enables the facile synthesis of a broad spectrum of chiral α-heterocyclic oxiranes, from mono- to tetrasubstituted variants but also advances the field of asymmetric epoxidation, opening pathways for the creation of complex chiral azaarenes.

**Table 1 | Summary of key reaction parameter effects[a]**

| Entry | Deviation from standard conditions | Yield (%)[b] | e.e. (%)[c] |
|---|---|---|---|
| 1 | None | 97 | 97 |
| 2 | (S)-**4b** | 77 | 93 |
| 3 | (S)-**4c** | 56 | 81 |
| 4 | (S)-**4d** | 89 | 95 |
| 5 | (S)-**4e** | 37 | 20 |
| 6 | (S)-**4f** | 57 | 67 |
| 7 | (S)-**4g** | 62 | 66 |
| 8 | without MgSO$_4$ | 53 | 97 |
| 9 | Na$_2$SO$_4$ instead of MgSO$_4$ | 45 | 81 |
| 10 | 4 Å MS instead of MgSO$_4$ | trace | ND |
| 11 | CHP instead of H$_2$O$_2$ | 16 | 93 |
| 12 | TBHP instead of H$_2$O$_2$ | 27 | 97 |
| 13[d] | CH$_2$Cl$_2$ instead of Toluene | 96 | 97 |

(S)-**4a**: Ar = 9-anthracenyl
(S)-**4b**: Ar = 1-napthyl
(S)-**4c**: Ar = 1-pyrenyl
(S)-**4d**: Ar = 2,4,6-$^i$Pr$_3$C$_6$H$_2$
(S)-**4e**: Ar = SiPh$_3$

(R)-**4f**
Ar = 9-anthracenyl

(R)-**4g**
Ar = 9-anthracenyl

[a]Standard reaction conditions using **1aa** (0.1 mmol), **2** (0.2 mmol), MgSO$_4$ (120 mg), and (S)–**4a** (3 mol %) at 35 °C for 5 h in toluene (1.0 mL) unless otherwise noted. [b]Isolated yields. [c]Enantiomeric excess (e.e.) and diastereomeric ratio (d.r.) values were determined using chiral HPLC analysis or $^1$H NMR analysis (GC). [d]18 h. *MS* molecular sieves; *CHP* cumene hydroperoxide; *TBHP tert*-butyl hydroperoxide; *ND* not detected.

## Results and Discussion

### Reaction Development

Initially, our research aimed to craft enantioenriched α-azaaryl oxiranes by devising a catalytic system that merges chiral phosphoric acids with hydrogen peroxide. The strategic insertion of a cyano group into alkenyl aza-heteroarene substrates significantly boosted both reactivity and stereoselectivity, thus streamlining the creation of essential α-azaaryl cyanoepoxides, instrumental for pharmaceutical innovations and subsequent chemical advancements. Detailed optimization of reaction parameters revealed ideal conditions for this epoxidation process, as summarized in Table 1. Notably, under optimal conditions (entry 1), the epoxidation of (Z)-configured substrate **1aa** with chiral phosphoric acid (S)−**4a**, hydrogen peroxide (30% *w/w* in H$_2$O), and MgSO$_4$ exclusively yielded *cis*-epoxide **3aa** with exceptional yield (97%) and enantioselectivity (97% e.e.). It is crucial to emphasize that no oxidation products involving the nitrogen atoms of aza-heteroarenes were detected, which highlights the specificity of the oxidation process towards the double bonds. A comprehensive screening of various phosphoric acid catalysts underscored the superior performance of the bis-anthracenyl-substituted catalyst (S)−**4a**, in terms of both activity and stereoselectivity. The reaction's efficiency was distinctly modulated by the choice of additives; in particular, the incorporation of MgSO$_4$ played

a critical role. Serving as an effective desiccant, MgSO$_4$ efficiently sequestered water−a byproduct inherent to the use of aqueous hydrogen peroxide−thereby enhancing the epoxidation process's efficiency[67,68]. The absence of MgSO$_4$ led to a discernible decrease in both yield and enantioselectivity, as evidenced by comparative analyzes presented in entries 8−10. Alternative oxidants, such as CHP and TBHP, were explored but did not enhance the product yield despite maintaining high enantioselectivity (entries 11 and 12). Solvent screening indicated that CH$_2$Cl$_2$, although slower in reaction rate, was as effective as toluene for this reaction (entry 13). This protocol has demonstrated robustness under varying conditions, exhibiting only slight reductions in enantioselectivity with increased temperatures (as documented in Supplementary Table 4) or higher substrate concentrations (detailed in Supplementary Table 5).

Upon refining our reaction protocols, we extended our investigations to encompass a broad spectrum of α-heterocyclic oxiranes with electron-withdrawing substituents, as demonstrated in Fig. 2. Efficient synthesis of these substrates was realized through Knoevenagel condensation, a process that involves combining heteroaryl acetonitrile with aldehydes, thereby enhancing the practicality and applicability of our method. Our exploration spanned an array of *N*-heterocyclic motifs, ranging from pyridine and diazine to triazine derivatives (**3aa**–**3af**), benzofused *N*-heterocycles (**3ag**–**3an**), azoles

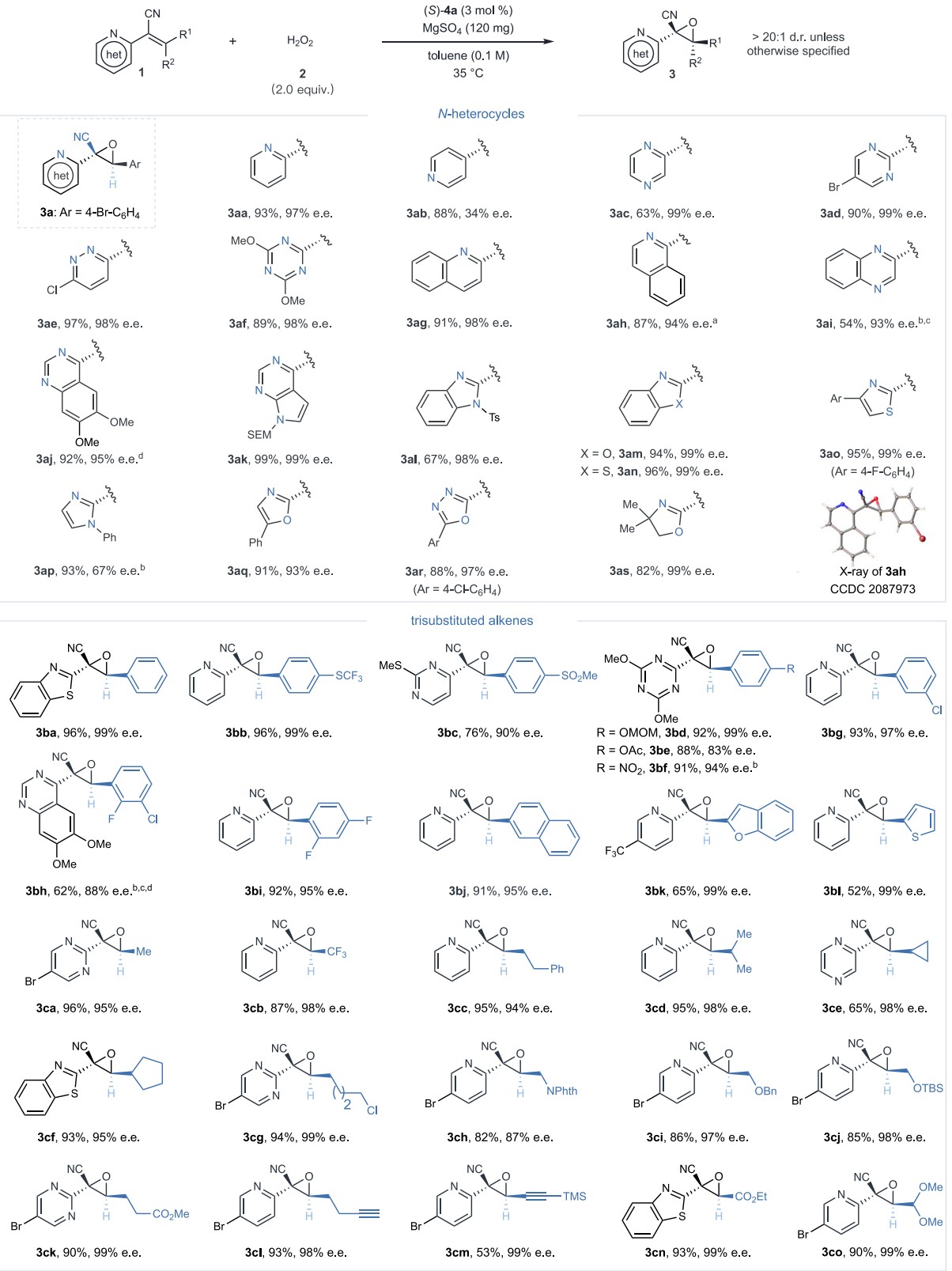

**Fig. 2 | Substrate scope for the synthesis of diverse α-heterocyclic oxiranes from various N-heterocycle substituted and trisubstituted alkenes.** Reaction conditions: (S)−**4a** (3 mol %), **1** (0.1 mmol), **2** (30% aq. H₂O₂, 0.2 mmol), MgSO₄ (120 mg) and toluene (1.0 mL) at 35 °C. Isolated yields were reported. Enantiomeric excess (e.e.) and diastereomeric ratio (d.r.) values were determined using chiral HPLC analysis or ¹H NMR spectroscopy. [a]Ar = 3-Br-C₆H₄. [b]Use of 5 mol % (S)−**4a**. [c]Reaction conducted at −20 °C. [d]Solvent mix of toluene/EtOAc in a 2/1 ratio. het, N-heterocycle; SEM, 2-(trimethylsilyl)ethoxymethyl; Ts, toluenesulfonyl; Mom, methoxymethyl; Ac, acetyl; Phth, phthalimidyl; Bn, benzyl; TBS, tert-butyl(di-methyl)silyl; TMS, trimethylsilyl.

(**3ao**−**3ar**), extending even to nonaromatics like oxazoline (**3as**), achieving consistently with excellent yields and selectivities. To evaluate the impact of the C=N bond's position within an azaarene, we performed the epoxidation of 4-pyridine derivatives efficiently, though this led to a modest decrease in enantioselectivity (**3ab**). The (2S,3R)-configuration of the epoxide products was assigned by comparison to product **3ah**, whose structure was unambiguously

established through X-ray crystallography analysis. Further investigations into β-position substituents of alkenyl aza-heteroarenes revealed their compatibility with a broad spectrum of aryl and heteroaryl groups, successfully leading to the formation of tri-substituted α-azaaryl oxiranes with superior enantioselectivities (**3ba**−**3bl**). In our pursuit of synthetically valuable transformations, we diversified our exploration by incorporating various aliphatic substituents at the β-

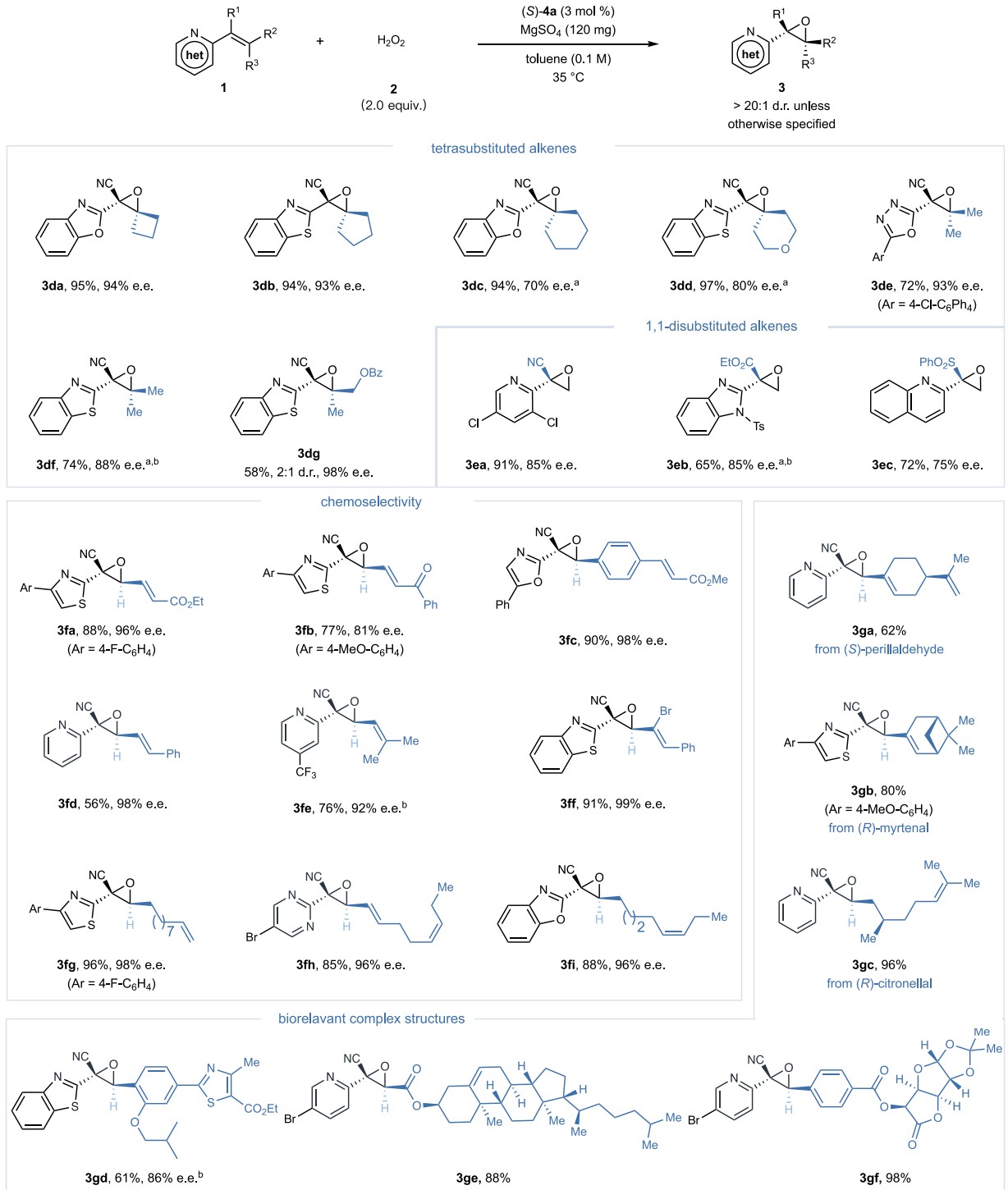

**Fig. 3 | Substrate scope including tetrasubstituted and 1,1-disubstituted alkenes, chemoselectivity, and biologically relevant complex structures.** Reaction conditions: (S)−**4a** (3 mol %), **1** (0.1 mmol), **2** (30% aq. H$_2$O$_2$, 0.2 mmol), MgSO$_4$ (120 mg) and toluene (1.0 mL) at 35 °C. Isolated yields were reported.

Enantiomeric excess (e.e.) and diastereomeric ratio (d.r.) values were determined using chiral HPLC analysis or [1]H NMR spectroscopy. [a]Reaction conducted at −20 °C. [b]Use of 5 mol % (S)−**4a**. het, N-heterocycle; Bz, benzoyl.

position of acrylonitriles. These substituents varied from linear and cyclic to conjugated forms and were supplemented with functional groups, including halide, phthalimide, ether, ester, alkyl alkyne, conjugated enyne, and acetal. This expansive substrate scope, conducted under optimized conditions, led to the generation of a series of alkylated oxiranes (3ca–3co) with noteworthy yields and enantiopurities, thus highlighting its broad utility in the synthesis of chiral α-azaaryl oxiranes.

Successfully navigating the steric complexities associated with tetrasubstituted alkenes, our approach delivers optically active oxiranes with varied substituents (cycloalkyl, dimethyl, asymmetric alkyl), reaching enantiomeric excesses up to 98% (3da–3dg), and highlights the method's potential in creating chiral quaternary stereocenters[69] (Fig. 3). The adaptability of our approach was further evidenced by its application to geminally disubstituted terminal alkenes, yielding key precursors for chiral tertiary alcohols (3ea–3ec). Despite a modest reduction in enantioselectivity (75–85% e.e.), this extension significantly broadens the method's applicability, accommodating a wider array of electron-withdrawing groups beyond cyano functionalities, including ester and sulfonyl groups. However, substituting electron-withdrawing groups with electron-donating methyl groups in substrates resulted in no detectable epoxide formation, underscoring a specific limitation related to the electronic effects of substituents, as illustrated in the Supplementary Fig. 69. Our protocol also exhibits unparalleled site-specificity, preferentially epoxidizing olefinic bonds adjacent to aza-heteroarenes across various substrate configurations. This specificity ensures consistent high yields and enantiomeric excesses, even in substrates featuring diverse alkene geometries and functionalities such as conjugated esters or ketones (3fa–3fi). Extending our method to naturally derived substrates, such as (S)-Perillaldehyde (3ga), (R)-Myrtenal (3gb), and (R)-Citronellal (3gc), alongside intermediates for Febuxostat (3gd) and derivatives of Cholesterol (3ge) and D-Glucuronic acid (3gf), we demonstrated its flexibility and efficiency in producing aza-heteroarenes with multiple stereocenters. The impressive diastereoselectivities achieved further affirm the technique's significant impact on the field of asymmetric epoxidation and its potential for intricate N-heterocycle modifications.

In our extended investigation into vinyl-substituted N-heteroaromatic compounds, as illustrated in Fig. 4, the original catalytic system utilizing (S)−4a demonstrated restricted efficacy. This diminished

**Fig. 4 | Range of mono-substituted and 1,2-disubstituted alkenyl azaarenes in the catalytic asymmetric epoxidation.** Reaction conditions: (R)−4j (3 mol %), 5 (0.1 mmol), 2 (30% aq. H₂O₂, 0.2 mmol), and toluene (1.0 mL) at 25 °C. Isolated yields were reported. Enantiomeric excess (e.e.) and diastereomeric ratio (d.r.) were determined using chiral HPLC analysis. ᵃUse of 3 mol % (S)−4a. ᵇUse of 5 mol % (R)−4j. Ts, toluenesulfonyl; TBS, tert-butyl(dimethyl)silyl.

effectiveness likely stems from the vinyl substrates' decreased propensity to stabilize an anionic charge at the α-position, especially compared to those with electron-withdrawing groups. To overcome this limitation, we adopted an innovative N-phosphinyl phosphoramide catalyst (R)−4j, distinguished by its additional basic P = O functionality[70], tailored to boost the epoxidation efficiency of these challenging substrates. Additionally, the strategic exclusion of MgSO₄ from the catalytic system resulted in substantial improvements in both reactivity and stereoselectivity, leading to the synthesis of diverse α-heterocyclic terminal epoxides (6a−6n) with high enantioselectivities (86−96% e.e.). The assignment of (R)-configuration to these terminal epoxides was confidently determined through a comparative analysis with 6b, the structure of which was unequivocally established via X-ray crystallography. Our methodology proved to be widely applicable, notably in the epoxidation of 1,2-disubstituted (E)-alkenes featuring quinoline, isoquinoline, benzoxazole and benzimidazole derivatives, consistently yielding the corresponding trans-epoxides (6o−6s) with yields and enantioselectivities ranging from 65−92%. Nonetheless, a notable limitation of our approach became apparent when attempting to epoxidize alkenyl pyridines possessing β-aryl groups (6t), which underscored certain methodological constraints. Moreover, our refined protocol exhibited commendable functional group tolerance, enabling the successful epoxidation of complex vinyl N-heteroarenes derived from compounds such as adenosine and sulfadoxine, and producing pharmaceutically significant oxiranes with high stereoselectivities (6u−6 v).

## Synthetic transformations

In an upscaled reaction, the epoxidation between 1ci and hydrogen peroxide 2 with a minimal catalyst concentration of 0.5 mol %, resulted in 2.8 g of oxirane 3ci, replicating the excellent performance of preliminary experiments by achieving an 85% yield and 97% e.e. (Fig. 5a). The versatility of α-cyanoepoxides such as 3cj and 3co, primarily attributed to the adaptable CN group, enables selective conversion into various derivatives including carboximidate (7a), carboximidic acid (7b), ketone (7c), ester (7d), and amine (7e), all while preserving the integrity of the oxirane structure and maintaining enantiomeric purity (Fig. 5b). The chiral oxiranes 3ci and 3cj also underwent regioselective and stereospecific ring-opening reactions, yielding compounds 7f−7h with vicinal stereocenters, where the site of reaction was determined by the choice of reagent. The formation of a five-membered ring compound 7i from 3co was achieved via acetonitrile-induced ring opening and subsequent cyclization, while 3ci was transformed into 7j through a piperidine-mediated rearrangement,

**Fig. 5 | Scale-up reaction and synthetic applications of the chiral products.** **a** Gram-scale synthesis of 3ci. Performed with 1ci (9.1 mmol) and (S)−4a (0.5 mol %) in toluene (0.1 M) at 35 °C for 7 h. **b** Derivatization of enantiopure α-heterocyclic cyanoepoxides. Conditions: (a) MeONa (1.2 equiv.), MeOH, rt, 12 h. (b) H₂O₂ (27.0 equiv.), Na₂CO₃ (3.0 equiv.), acetone, rt, 12 h. (c) TMSCl (3.0 equiv.), p-MeO-PhMgBr (2.0 equiv.), toluene, −40 °C, 3 h. (d) BF₃·Et₂O (3.0 equiv.), CH₂Cl₂, 40 °C, 12 h. (e) DIBAL-H (5.0 equiv.), toluene, −78 °C, 3 h, then di-tert-butyl dicarbonate (2.0 equiv.), NaHCO₃ (2.0 equiv.), MeOH, rt, 24 h. (f) DIBAL-H (5.0 equiv.), toluene, −78 °C, 3 h, followed by di-tert-butyl dicarbonate (2.0 equiv.), NaHCO₃ (2.0 equiv.), MeOH, rt, 24 h, then Zinc chloride (1.5 equiv.), CH₂Cl₂, 0 °C, 2 h. (g) Zinc chloride (1.5 equiv.), acetyl chloride (2.0 equiv.), CH₂Cl₂, 35 °C, 12 h. (h) TEAB (2.0 equiv.), BF₃·Et₂O (2.0 equiv.), CH₂Cl₂, 0 °C, 1 h. (i) CH₃CN (1.5 equiv.), KHMDS (3.0 equiv.), THF, −20 °C, 2 h. (j) piperidine (3.0 equiv.), EtOH, 50 °C, 18 h. Isolated yields were reported. Enantiomeric excess (e.e.) and diastereomeric ratio (d.r.) values were determined using chiral HPLC analysis or gas chromatography (GC). The absolute stereochemistry of compound 7d was assigned as (1 S,5 S)-configuration via X-ray crystallography analysis. TBS, tert-butyl(dimethyl)silyl; DIBAL-H, diisobutylaluminium hydride; TEAB, tetraethylammonium bromide; KHMDS, potassium bis(-trimethylsilyl)amide.

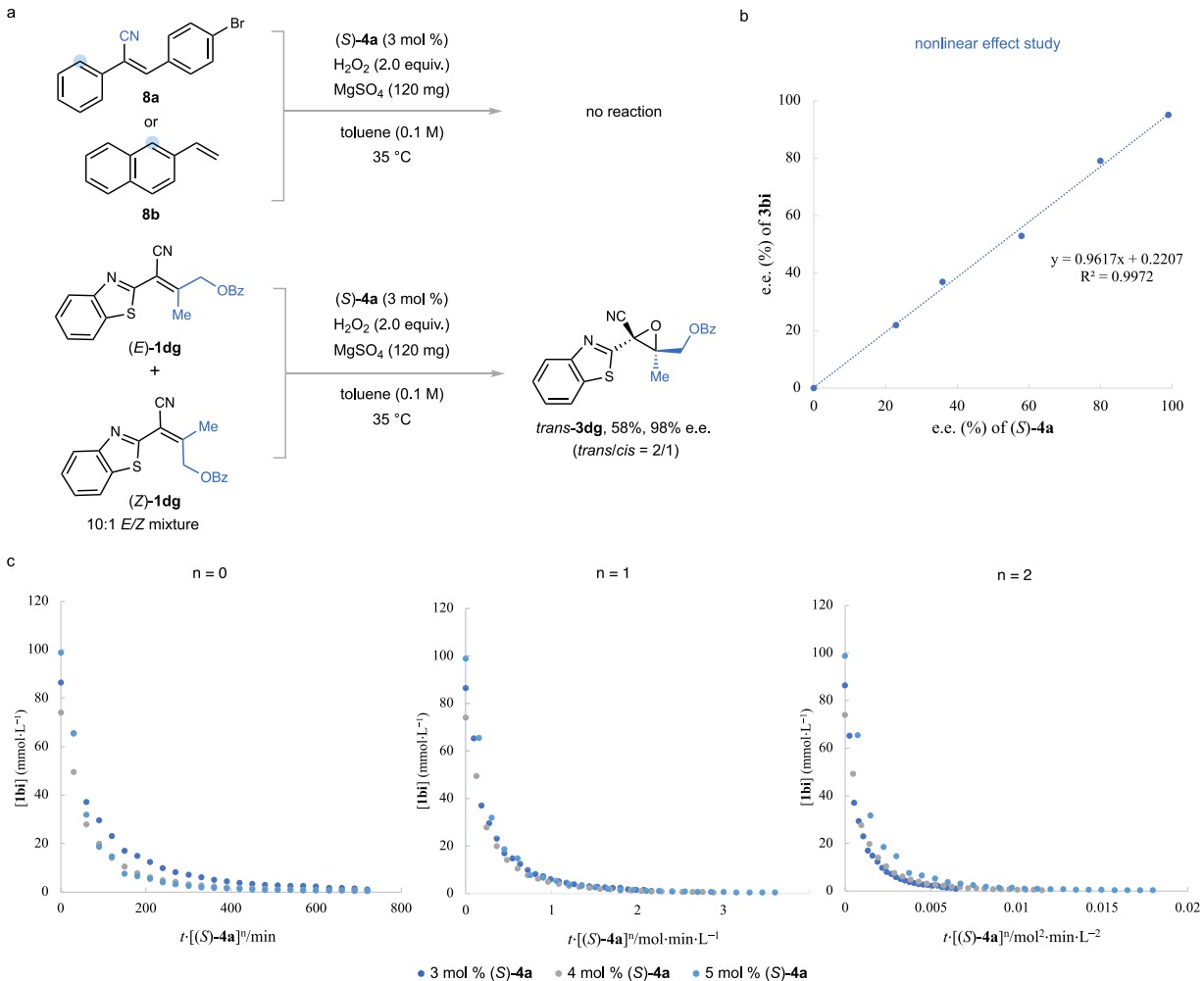

**Fig. 6 | Mechanistic insights into the epoxidation reaction. a** Reactivity comparison between compounds **8a** and **8b** under optimized conditions. **b** Study of the nonlinear effect. **c** Determination of reaction order by variable time normalization analysis following the method of Burés[71]: a first-order dependence on the catalyst concentration with product concentration plotted against a normalized time scale.

The convergence of plots from reactions with varying catalyst concentrations, where the exponent $n$ equals the catalyst order ($n = 1$), signifies first-order kinetics. Conversions of **1bi** were quantified by gas chromatography (GC), employing *n*-dodecane as the internal standard.

albeit in moderate yields. These transformations illustrate the varied reactivity of chiral α-heterocyclic oxiranes, establish an approach for the catalytic stereoselective synthesis of densely functionalized aromatic *N*-heterocycles adorned with vicinal stereocenters.

## Mechanistic investigations

Through a comprehensive mechanistic investigation, our control experiments shed light on the substrate specificity of the epoxidation process. The critical importance of the nitrogen atom in azaarenes was established that its removal under otherwise optimal conditions halted the epoxidation process, thereby affirming the essential role of the C = N bond as an activation site (Fig. 6a). By epoxidizing a 10:1 *E/Z* mixture of compound **1dg**, we observed the formation of the *trans/cis* stereoisomers **3dg** at a diminished ratio of 2:1, illustrating a compromise in stereochemical integrity. Additionally, a control experiment—excluding hydrogen peroxide but including chiral phosphoric acid—altered the *E/Z* ratio of **1dg** from 10:1 to 2.5:1, indicating the profound influence of catalyst-substrate interactions in isomerization processes. In addition, a linear relationship between the enantioselectivities of the catalyst and the epoxide product **3bi** was observed during epoxidation, suggesting the influence of a monomeric catalyst in the reaction's crucial transition state (Fig. 6b). Kinetic studies, based on Burés' graphical

method[71], confirmed the reaction's first-order dependency on catalyst concentration, accentuating the catalyst's vital role in shaping both the stereooutcome and the overall reaction kinetics (Fig. 6c).

To decode the molecular basis of the observed stereoselectivity, we employed density functional theory (DFT) calculations, focusing on **1aa** as the model compound. The computational insights unveiled the formation of a pivotal intermediate **Int1-SR**, stabilized by a combination of two hydrogen bonds and one O–H⋯π interaction involving the catalyst (*S*)-**4a**, hydrogen peroxide, and **1aa**. This stabilization manifests as a substantial energy reduction of −19.6 kcal/mol (Fig. 7a), demonstrating the essential role of non-covalent interactions (NCI) in steering the reaction pathway toward high stereoselectivity. Progressing from **Int1-SR** to the final epoxide necessitates overcoming a free energy barrier of 20.4 kcal/mol at the transition state **TS1-SR**. In a comparative analysis, the non-cyano variant of **1aa** exhibited a notably higher barrier of 27.6 kcal/mol, illustrating the profound effect of the cyano substituent on reducing the energy requirements of the reaction. This finding underscores the significant impact of substituent nature on catalytic efficiency and stereoselectivity, effectively facilitating ring closure via **TS2-SR** and completing the epoxidation cycle.

Distinguishing **TS1-SR** from its enantiomeric counterpart, **TS1-RS**, by a Gibbs free energy difference of 4.0 kcal/mol, directly correlates with the experimentally observed enantiomeric excess of 97% e.e.,

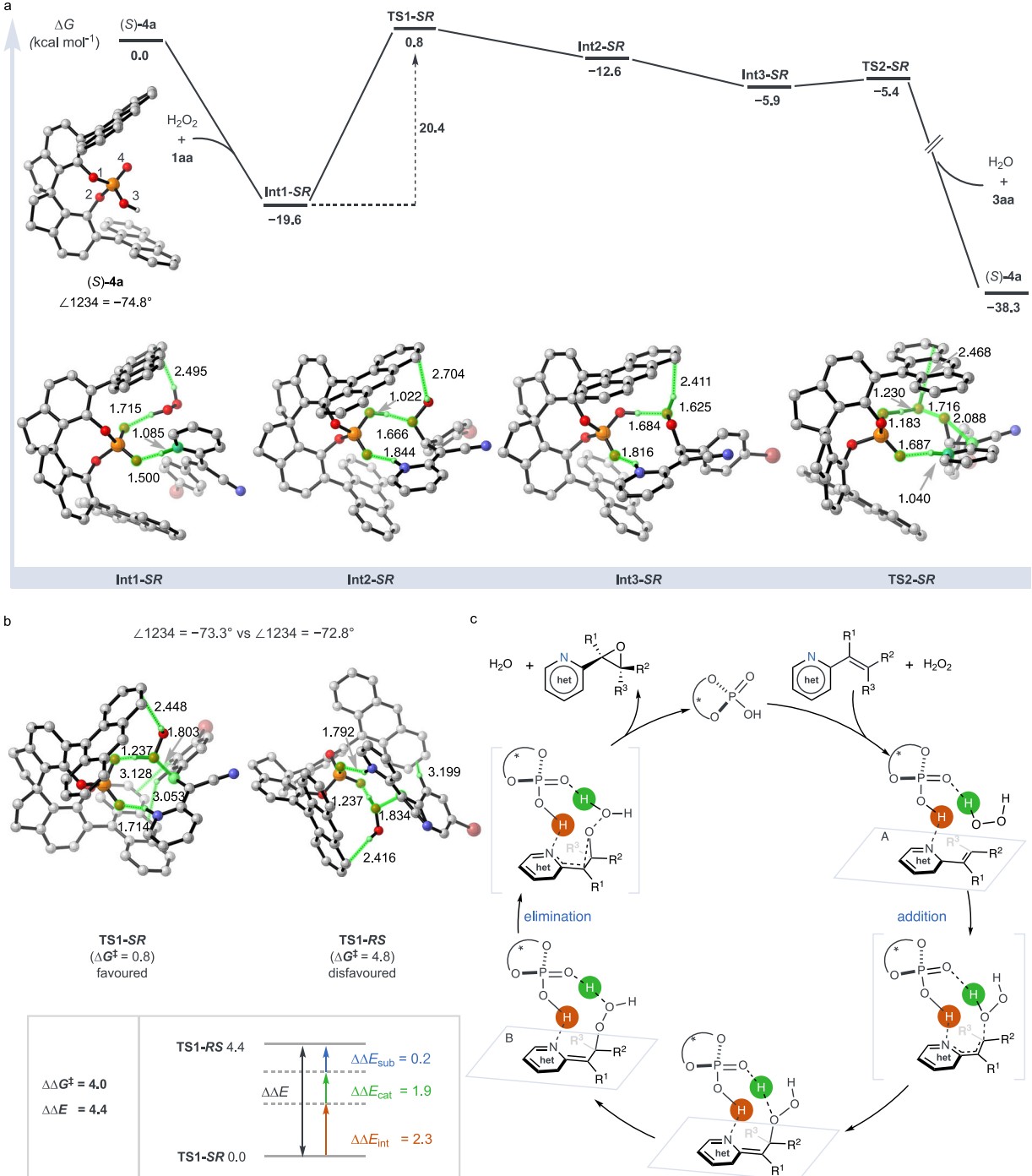

**Fig. 7 | Computational studies and proposed mechanisms. a** Possible reaction pathways for the asymmetric epoxidation of alkenyl *N*-heteroarene **1aa** with hydrogen peroxide catalyzed by (*S*)−**4a**. The relative Gibbs free energies and bond distances are given in kcal mol⁻¹ and Å, respectively. The dihedral angles of the catalyst centers in (*S*)−**4a** are represented as ∠1234. **b** Density Functional Theory (DFT)-optimized structures of transition states **TS1-SR** and **TS1-RS**. **c**, Proposed catalytic cycle.

providing a computational validation of the reaction's stereoselectivity (Fig. 7b). An energy decomposition analysis further elucidated a significant electronic energy difference (ΔΔE) of 4.4 kcal/mol between the transition states **TS1-SR** and **TS1-RS**. This discrepancy is primarily attributed to the differential distortion of the catalyst and its nuanced interactions with the substrate. Notably, the structural analysis highlighted minimal substrate differences between these states, emphasizing the critical role of the catalyst's conformational dynamics in stereoselectivity. Specifically, we observed dihedral angle shifts from −74.8° in the resting state to −73.3° in **TS1-SR** and further to −72.8° in

**TS1-RS**, underscoring the importance of these conformational changes for achieving high enantioselectivity. Furthermore, **TS1-SR** exhibits a 2.3 kcal/mol greater interaction energy with the substrate than **TS1-RS**, indicative of stronger non-covalent bonding that reinforces the observed ΔΔE$_{int}$ value, as shown by NCI plots in Supplementary Fig. 117.

Integrating data from both detailed experimental studies and DFT calculations, we propose a catalytic mechanism for the stereoselective epoxidation of alkenyl aza-heteroarenes, as depicted in Fig. 7c. Initiation of this catalytic cycle involves the chiral phosphoric acid-mediated

simultaneous activation of both the aza-heteroarene substrates and hydrogen peroxide, primarily through electrostatic and hydrogen-bonding interactions. This dual activation gives rise to complex **A**, wherein the aza-heteroarene substrate is protonated by chiral phosphoric acid and intimately associated with hydrogen peroxide. The ensuing stage involves a nucleophilic attack by hydrogen peroxide on the activated alkene, accompanied by an essential proton transfer, leading to the formation of the enantioenriched complex **B**. The culmination of this sequence is achieved through a stereospecific elimination step, effectively reinstating the aromaticity of the aza-heteroarene and resulting in the synthesis of the desired chiral oxirane with notable enantiopurity.

In summary, our research pioneers an organocatalytic approach for the enantioselective epoxidation of alkenyl aza-heteroarenes, leveraging hydrogen peroxide as an environmentally benign oxidant in concert with chiral Brønsted acid catalysis. The key to this advancement is the synergistic interplay of chiral phosphoric acid with C = N functionalities embedded within azaarenes, markedly enhancing both the selectivity and reactivity of the epoxidation process. Our approach has been demonstrated to be highly effective across a diverse range of substrates, achieving enantioselectivities of up to 99% e.e. and exhibiting specific reactivity towards targeted sites, thereby enabling the precise creation of chiral azaaryl molecules with consecutive stereocenters. Kinetic studies and DFT calculations elucidate the reaction's stereoselectivity, highlighting the importance of the chiral phosphoric acid in orchestrating the epoxidation process through a finely tuned balance of non-covalent interactions. These findings contribute to a deeper understanding of the mechanistic underpinnings of asymmetric epoxidation. This advance not only marks a crucial development in asymmetric epoxidation but also provides a robust, versatile platform for constructing complex *N*-heterocyclic frameworks, offering extensive utility in both pharmaceutical development and synthetic chemistry.

## Methods

### General procedure for the asymmetric epoxidation of tri-, tetra- and 1,1-disubstituted alkenyl aza-heteroarenes

A mixture of alkenyl aza-heteroarenes (0.1 mmol), (*S*)−**4a** (3 mg, 0.03 mmol), and $MgSO_4$ (120 mg) was placed in a 4 mL vial with a magnetic stir bar. Freshly distilled toluene (1.0 mL) was added, followed by a dropwise addition of $H_2O_2$ (30% *w/w* in $H_2O$, 16 μL, 0.2 mmol). The mixture was stirred at 35 °C and its progress monitored via TLC. Once the reaction was deemed complete, the solution was filtered through a short celite pad. The combined organic layer was concentrated under reduced pressure and further purified via flash column chromatography on silica gel to isolate the desired products.

### General procedure for the asymmetric epoxidation of mono-substituted and 1,2-disubstituted alkenyl aza-heteroarenes

A mixture of alkenyl aza-heteroarenes (0.1 mmol), (*R*)−**4j** (4 mg, 0.03 mmol) was placed in a 4 mL vial with a magnetic stir bar. Freshly distilled toluene (1.0 mL) was added, followed by a dropwise addition of $H_2O_2$ (30% *w/w* in $H_2O$, 16 μL, 0.2 mmol). The mixture was stirred at 25 °C and its progress was monitored via TLC. Once the reaction was deemed complete, the solution was filtered through a short celite pad. The combined organic layer was concentrated under reduced pressure and further purified via flash column chromatography on silica gel to isolate the desired products.

## Data availability

The crystallographic data generated in this study have been deposited in the Cambridge Crystallographic Data Center (CCDC) under deposition numbers CCDC 2087974 (**1ah**), 2087973 (**3ah**), 2330838 (**6b**), and 2299941 (**7d**). Copies of the data can be obtained free of charge via www.ccdc.cam.ac.uk. The cartesian coordinates are available in Excel format as source data. All other data supporting the findings of this study, including experimental procedures and compound characterization, NMR, HPLC, and X-ray analyses, are available within the article and its Supplementary Information or from the corresponding authors. Source data are provided in this paper.

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

## Acknowledgements

Bin Mao acknowledges funding supported by Zhejiang University of Technology (H4148220383). Shao-Fei Ni acknowledges funding supported by the open research fund of Songshan Lake Materials Laboratory (2023SLABFN16) and the STU Scientific Research Foundation for Talents (NTF20022). Bin Mao thanks Prof. Dr. Benjamin List and Prof. Dr. Yixia Jia for their helpful suggestions. Bin Mao acknowledges funding supported by Zhejiang University of Technology (KYY-HX-20220383).

## Author contributions

B.M. conceived and directed the project. H.W., W.C., and M.L. designed and conducted the synthetic experiments. A.F. and S.X. prepared the substrates for reaction scope evaluation. C.M. and S.N. performed computational studies. B.M. prepared this manuscript with feedback from J.W. and Y.Z.

## Competing interests

The authors declare no competing interests.
