## [Peer Review File · Nature Communications]

Chiral Phosphoric Acid-Catalyzed Asymmetric Epoxidation of Alkenyl Aza-Heteroarenes Using Hydrogen PeroxideREVIEWER COMMENTS

Reviewer #1 (Remarks to the Author):

See the attachment for further information.

Reviewer #2 (Remarks to the Author):

The manuscript describes a novel organocatalyzed approach for CPA-catalyzed asymmetric epoxidation of alkenyl aza-heteroarenes using H₂O₂ as the oxidant. This method relies on a synergistic blend of electrostatic and hydrogen-bonding interactions, enabling the simultaneous activation of the aza-heteroarenes and H₂O₂ and resulting in azaaryl molecules with high enantioselectivity. This work is an important piece of study. The reaction scope is studied as well, and a wide range of chiral azaaryl molecules are obtained. Nevertheless, to meet the high standard of Nature Communications, there are some issues with this manuscript. Therefore, this manuscript needs minor revisions.

- 1) The additive MgSO₄ is necessary to the transformation. Some comments or references should be provided to account for the influence of MgSO₄ on the transformation.
- 2) In the part of substrate scope, it is interesting that some products were obtained in low yields, such as 3ai, 3bl, and 3cm, it would be great if the author provided some explanations.
- 3) The asymmetric epoxidation of tri-, tetra- and 1,1-disubstituted alkenyl aza-heteroarenes have been investigated carefully, however, only electron-withdrawing groups, including cyano, ester and sulfonyl groups, were exhibited in the manuscript. How about electron-donating groups? This will directly affect the universality of the current reporting methods.
- 4) H₂O has always existed in the reaction system. Have the authors observed the products of epoxide hydrolysis?

Reviewer #3 (Remarks to the Author):

This is a study on the catalytic asymmetric epoxidation of α -heteroaryl β -substituted acrylonitriles. Using a chiral phosphoric acid catalyst, the epoxidation reaction was promoted to give the products with enantioselective manner. Although the working hypothesis of the hydrogen-bonding activation of α -heteroaryl substituents are interesting, the field of asymmetric epoxidation of electron-deficient alkenes has been well established. The current manuscript is missing in the novelty and urgency required for the Nature Communications. For submitting to the other specialized journal, the title, abstract, and the introduction should be reconsidered. This isn't the chemistry of epoxidation of the simple "Alkenyl Aza-Heteroarenes". This is the epoxidation of electron-deficient acrylonitriles. The references for the recent reports on the catalytic asymmetric epoxidation of acrylonitriles should be updated.

Reviewer #4 (Remarks to the Author):

This work realizes a high enantioselective synthesis of α -azaheteroaryl oxiranes with moderate to good yields employing the chiral phosphoric catalysts. In this manuscript, the first-order dependency of the catalyst concentration was observed by kinetic studies. Furthermore, the authors investigate the underlying mechanism of stereoselectivity and provide the energy profile of the proposed reaction pathway through DFT calculations. Mechanistically, the chiral phosphoric simultaneously activates the substrate and hydrogen peroxide through the synergistic action of hydrogen bond and electrostatic interactions. From the perspective of application requirements, various derivatives synthesized from α -azaheteroaryl oxiranes demonstrate the potential application value of α -azaheteroaryl oxiranes as a chemical precursor. Altogether, this work is novel but a minor revision is required before published.

1. In Table 1, the presence of MgSO_4 significantly affects the yield of 3aa. If possible, the authors should explain the role of MgSO_4 in this synthesis.
2. As shown in the manuscript, the structure of 6a was established via X-ray crystallography. But only the X-ray structure of 6b was shown in Figure 3. On the other hand, there is only one chiral atom in 6a, 6a was shown as (R)-configuration instead of (S)-configuration. The authors should check the crystal data provided in SI.
3. In Figure 3, the authors completed the synthesis of α -heterocyclic terminal epoxides catalyzed by (R)-4j from the vinyl-substituted N-heteroaromatic compounds without CN. In SI, the authors provide the cartesian coordinates of TS1-SR-H and Int1-SR-H without CN. If the mechanism of the reaction pathway shown in Figure 3 has been investigated through DFT calculations, it's better to provide the energy profile of the synthesis pathway shown in Figure 3.
4. In Figure 5c, when $n=2$, the unit of $t\cdot[(S)\text{-}4a]_n$ is $\text{mol}^2\cdot\text{min}\cdot\text{L}^{-2}$ instead of $\text{mol}\cdot\text{min}\cdot\text{L}^{-2}$. Please check the unit in the manuscript again.

This manuscript utilizes chiral phosphoric acid as the catalyst for the enantioselective epoxidation of alkenyl aza-heteroarenes using environmentally friendly hydrogen peroxide. This method addresses the challenge of the asymmetric epoxidation of aza-alkenes, especially in achieving high stereoselectivity and chemoselectivity. Through this approach, researchers were able to efficiently synthesize a series of chiral α -aza-heteroaryl epoxides, which is of significant importance for the preparation of complex chiral drugs and other functional molecules. The article also reveals the mechanism behind stereoselectivity through kinetic studies and density functional theory (DFT) calculations, emphasizing the crucial role of chiral phosphoric acid in guiding this complex asymmetric epoxidation process. However, the authors need to address the following issues.

- 1) When hydrogen peroxide oxidizes the double bonds of aza-heteroarenes, have the authors observed products resulting from the oxidation of the nitrogen atom in the aza-heteroarenes? Please describe this in the text.
- 2) Reference 45 is not related to enantioselective addition work. Please carefully check all references for similar issues. Additionally, for the photoredox catalysis of aza-arenes involved in radical reactions, some of the latest literature should be supplemented. For example: *J. Am. Chem. Soc.* **2021**, *143*, 4024–4031; *Angew. Chem. Int. Ed.* **2022**, *61*, e202115110; *Chin. J. Catal.* **2022**, *43*, 558–563; *J. Am. Chem. Soc.* **2023**, *145*, 18307–18315.
- 3) For the study of non-linear effects, please provide the original HPLC charts, rather than just supplying the ee values.
- 4) Many ^1H NMR spectra are not clean. Please provide clean ^1H NMR charts after further purification, for example: compounds 3ab, 3ac, 3ad, 3af, etc.

After making revisions based on the above comments, the reviewer believes it is suitable for publication in Nature Communications.

Responses to the comments of reviewers

Reviewer 1

Comment 1: When hydrogen peroxide oxidizes the double bonds of aza-heteroarenes, have the authors observed products resulting from the oxidation of the nitrogen atom in the aza-heteroarenes? Please describe this in the text.

Response: *Thank you for your insightful query. In response to your comment, we have carefully examined our experimental data and did not observe any products that could be attributed to the oxidation of the nitrogen atom in the aza-heteroarenes. We have clarified this point in the revised manuscript by adding the following sentence: " It is crucial to emphasize that no oxidation products involving the nitrogen atoms of aza-heteroarenes were detected, which highlights the specificity of the oxidation process towards the double bonds."*

Comment 2: Reference 45 is not related to enantioselective addition work. Please carefully check all references for similar issues. Additionally, for the photoredox catalysis of aza-arenes involved in radical reactions, some of the latest literature should be supplemented. For example: J. Am. Chem. Soc. 2021, 143, 4024–4031; Angew. Chem. Int. Ed. 2022, 61, e202115110; Chin. J. Catal. 2022, 43, 558–563; J. Am. Chem. Soc. 2023, 145, 18307–18315.

Response: *Thank you for pointing out the discrepancy with Reference 45 and for suggesting additional literature. Upon careful review, we have removed Reference 45 as it indeed did not pertain to enantioselective addition work. We have meticulously checked all other references to ensure their relevance and accuracy within the context of our study. Furthermore, we have added the recommended articles on the photoredox catalysis of aza-arenes involved in radical reactions to our literature review to enhance our discussion and support our methodology. These additions include:*

52. Yin, Y. et al. Conjugate addition–enantioselective protonation of N-aryl glycines to α -branched 2-vinylazaarenes via cooperative photoredox and asymmetric catalysis. J. Am. Chem. Soc. 140, 6083–6087 (2018).

53. Cao, K. et al. Catalytic enantioselective addition of prochiral radicals to vinylpyridines. J. Am. Chem. Soc. 141, 5437–5443 (2019).

54. Yin, Y. et al. All-carbon quaternary stereocenters α to azaarenes via radical-based asymmetric olefin difunctionalization. J. Am. Chem. Soc. 142, 19451–19456 (2020).

55. Kong, M. et al. *Catalytic reductive cross coupling and enantioselective protonation of olefins to construct remote stereocenters for azaarenes*. *J. Am. Chem. Soc.* 143, 4024–4031 (2021).

Comment 3: For the study of non-linear effects, please provide the original HPLC charts, rather than just supplying the ee values.

Response: *Thank you for your request to include the original HPLC charts for our study of non-linear effects. In response to your comment, we have added the requested HPLC charts to the supplementary information, which can be found on pages 133-135 of the supplementary file. Furthermore, we repeated the experiments to ensure accuracy and have updated the ee values for the products, which now stand at 95%. These revised values and the corresponding HPLC charts have been replaced in the manuscript to reflect these changes accurately. We believe that these additions and corrections will provide a clearer understanding of the data and enhance the integrity of our reported findings.*

Comment 4: Many ¹H NMR spectra are not clean. Please provide clean ¹H NMR charts after further purification, for example: compounds 3ab, 3ac, 3ad, 3af, etc.

Response: *Thank you for highlighting the issue with the clarity of the ¹H NMR spectra for several compounds in our manuscript. We have undertaken additional purification for the mentioned compounds and have obtained cleaner ¹H NMR spectra. These updated spectra can be found in the supplementary information:*

Compound 3ab: page S259

Compound 3ac: page S260

Compound 3ad: page S261

Compound 3af: page S263

In addition, we have also updated the spectra for compounds 1ad (page S177) and 1an (page S187), as well as for compound 3ea (page S319). We trust that these updates will better demonstrate the purity of the compounds and support the integrity of our data.

Reviewer 2

Comment 1: The additive MgSO₄ is necessary to the transformation. Some comments or references should be provided to account for the influence of MgSO₄ on the transformation.

Response: *We acknowledge the importance of elucidating the role of MgSO₄ in our catalytic system and thank you for the opportunity to clarify its significance. The data we have gathered, detailed in the supplementary information (Table S2.3), clearly indicates that MgSO₄ is essential for achieving optimal yields and reaction efficiencies. MgSO₄ acts as a desiccant, mitigating the deleterious effects of water, which is both a constituent of the aqueous hydrogen peroxide solution and a byproduct of the epoxidation reaction. This removal of water by MgSO₄ is critical in driving the reaction forward to the desired oxiranes with high yield and selectivity. The transformative impact of MgSO₄ on our system is supported by analogous findings in the literature:*

67 Liao, S. et al. Activation of H₂O₂ by chiral confined Brønsted acids: a highly enantioselective catalytic sulfoxidation. J. Am. Chem. Soc. 134, 10765–10768 (2012).

68 Ma, L. et al. Chiral Brønsted-acid-catalyzed asymmetric oxidation of sulfenamide by using H₂O₂: a versatile access to sulfinamide and sulfoxide with high enantioselectivity. ACS Catal. 9, 1525–1530 (2019).

We have amended the manuscript to reflect this discussion, incorporating the revised statement: “The reaction's efficiency was distinctly modulated by the choice of additives; in particular, the incorporation of MgSO₄ played a critical role. Serving as an effective desiccant, MgSO₄ efficiently sequestered water—a byproduct inherent to the use of aqueous hydrogen peroxide—thereby enhancing the epoxidation process's efficiency. The absence of MgSO₄ led to a discernible decrease in both yield and enantioselectivity, as evidenced by comparative analyses presented in entries 8–10.” This amendment and the pertinent references collectively offer a comprehensive overview of the influence of MgSO₄ in our catalytic system.

Comment 2: In the part of substrate scope, it is interesting that some products were obtained in low yields, such as 3ai, 3bl, and 3cm, it would be great if the author provided some explanations.

Response: *Upon repeating the reactions for a more detailed analysis, we found that the transformations involving substrates such as 1ai and 1bl were highly efficient, leading to complete consumption of the starting materials. Despite the clean and complete reactions, the isolated yields for these compounds were lower than anticipated. Our investigation has shown that these lower yields primarily resulted from losses incurred during the chromatographic purification step. This*

purification is crucial for achieving the high purity required for structural analysis and further applications, yet it appears to lead to significant product loss.

For the conjugated enyne substrate 1cm, we implemented an optimized purification protocol and extended the reaction time to 48 hours, which improved the yield from 31% to 53%. Although the starting material was not fully consumed, this adjustment provided a better yield compared to initial attempts. This adjustment, while enhancing the yield, also highlighted that the starting material was not fully consumed, indicating variable reaction kinetics for this particular substrate. This case exemplifies the need for tailored approaches for each substrate to balance reaction conditions and purification methods effectively.

We are grateful for your insightful comment, as it prompted a deeper investigation into these aspects of our research, ultimately enhancing the quality and depth of our study.

Comment 3: The asymmetric epoxidation of tri-, tetra- and 1,1-disubstituted alkenyl aza-heteroarenes have been investigated carefully, however, only electron-withdrawing groups, including cyano, ester and sulfonyl groups, were exhibited in the manuscript. How about electron-donating groups? This will directly affect the universality of the current reporting methods.

Response: *We appreciate your inquiry regarding the incorporation of electron-donating groups within our asymmetric epoxidation studies. In pursuit of addressing this point, we extended our experiments to include substrates like 1,1-disubstituted alkenyl aza-heteroarenes, where typical electron-withdrawing groups were replaced by methyl groups—an electron-donating substituent. Despite applying the same epoxidation conditions optimized for electron-withdrawing groups, no epoxide formation was observed in these modified substrates (see SI, page S104).*

This result clearly indicates that electron-donating groups significantly hinder the reaction process, affecting the formation of the desired epoxide products. To clarify this limitation, we have included a specific discussion in our manuscript: "However, substituting electron-withdrawing groups with electron-donating methyl groups in substrates resulted in no detectable epoxide formation (see Supplementary Information for details), underscoring a specific limitation related to the electronic effects of substituents." This addition aims to enhance the understanding of our methodology's scope and the electronic influence of substituents on the reaction efficacy. We believe this insight is crucial for further refining the protocol and understanding the mechanistic aspects of our catalytic system.

Comment 4: H₂O has always existed in the reaction system. Have the authors observed the products of epoxide hydrolysis?

Response: *Thank you for highlighting the potential for epoxide hydrolysis in our reaction system, given the presence of water. Upon careful analysis of the reaction mixtures, we did not observe any products indicative of epoxide hydrolysis. This observation suggests that under our experimental conditions, the epoxides remain stable, and hydrolysis does not significantly compete with the desired reaction pathways.*

Reviewer 3

Comment : This is a study on the catalytic asymmetric epoxidation of α -heteroaryl β -substituted acrylonitriles. Using a chiral phosphoric acid catalyst, the epoxidation reaction was promoted to give the products with enantioselective manner. Although the working hypothesis of the hydrogen-bonding activation of α -heteroaryl substituents are interesting, the field of asymmetric epoxidation of electron-deficient alkenes has been well established. The current manuscript is missing in the novelty and urgency required for the Nature Communications. For submitting to the other specialized journal, the title, abstract, and the introduction should be reconsidered. This isn't the chemistry of epoxidation of the simple "Alkenyl Aza-Heteroarenes". This is the epoxidation of electron-deficient acrylonitriles. The references for the recent reports on the catalytic asymmetric epoxidation of acrylonitriles should be updated.

Response: *Thank you for your insightful critique and the opportunity to clarify and emphasize the unique contributions of our work within the field of asymmetric epoxidation. We value your suggestions to refine our presentation and have made significant updates to reinforce the manuscript's alignment with the high standards of Nature Communications.*

While we appreciate the well-established nature of asymmetric epoxidation of electron-deficient alkenes, our study indeed goes beyond simple electron-deficient acrylonitriles. Our research introduces a novel chiral phosphoric acid catalyst that has been uniquely optimized for α -heteroaryl β -substituted acrylonitriles—a substrate class that has not been extensively explored, particularly in the context of pharmaceutical applications. Moreover, our study expands beyond traditional electron-deficient acrylonitriles to include substrates with diverse electron-withdrawing groups such as esters and sulfonyl groups. This not only broadens the scope but also

enhances the applicability of our methodology, addressing a wider array of chemical frameworks than previously reported.

Additionally, as highlighted in Figure 3 of our manuscript, our study encompasses a range of mono-substituted and 1,2-disubstituted alkenyl azaarenes, including terminal alkenes which do not conform to the typical definition of electron-deficient acrylonitriles. This extensive substrate range underlines the versatility and innovative nature of our approach, effectively overcoming the steric and electronic limitations that hamper existing methods.

In response to your comments, we have meticulously updated our literature review to include the most recent advancements in asymmetric epoxidation, particularly those targeting acrylonitriles and similar substrates. Noteworthy among these are:

*33. Volpe, V. et al. Catalytic enantioselective access to dihydroquinoxalinones via formal α -halo acyl halide synthon in one pot. *Angew. Chem. Int. Ed.* 60, 23819–23826 (2021).*

*34. Ogino, E., Nakamura, A., Kuwano, S. & Arai, T. Chiral C2-symmetric aminomethylbinaphthol as synergistic catalyst for asymmetric epoxidation of alkylidenemalononitriles: easy access to chiral spirooxindoles. *Org. Lett.* 23, 1980–1985 (2021).*

*35. Ogino, E., Kuwano, S. & Arai, T. Chiral aminomethylbinaphthol-catalyzed diastereo- and enantioselective epoxidation of trisubstituted acrylonitriles. *Adv. Synth. Catal.* 364, 1503 – 1506 (2022).*

These citations not only contextualize our contributions but also clearly delineate how our findings address previously unmet challenges, providing robust support for the novelty and applicability of our approach.

We believe the revisions and additions to our manuscript thoroughly address the concerns raised and convincingly demonstrate the novelty and significance of our research. By pushing the boundaries of asymmetric epoxidation to include complex aza-heterocyclic frameworks, our work opens new avenues for the development of pharmaceutically relevant compounds and fills a critical gap in the synthetic methodology. Thank you again for your review and valuable feedback, which have undoubtedly strengthened our manuscript.

Reviewer 4

Comment 1: In Table 1, the presence of MgSO₄ significantly affects the yield of 3aa. If possible, the authors should explain the role of MgSO₄ in this synthesis.

Response: *Thank you for your comment regarding the role of MgSO₄ in our synthesis, which indeed plays a pivotal role in the reaction outcomes as highlighted in Table 1. As we have discussed in response to a similar query from Reviewer 2, MgSO₄ serves a crucial function in our catalytic system.*

MgSO₄ acts primarily as a desiccant, absorbing water that is present both as a constituent of the aqueous hydrogen peroxide solution and as a byproduct formed during the epoxidation reaction. This absorption is essential for minimizing the hydrolytic degradation of the epoxide product, thereby promoting higher yields and enhanced reaction efficiency. The absence of MgSO₄ significantly reduces both the yield and enantioselectivity of the reaction, as demonstrated in the comparative analyses presented in entries 8–10 of Table 1.

For detailed insights into the specific effects of MgSO₄ and a more comprehensive discussion of its role, please refer to the supplementary information (Table S2.3) and the related discussion sections in our manuscript.

We have amended the manuscript to reflect this discussion, incorporating the revised statement: “The reaction's efficiency was distinctly modulated by the choice of additives; in particular, the incorporation of MgSO₄ played a critical role. Serving as an effective desiccant, MgSO₄ efficiently sequestered water—a byproduct inherent to the use of aqueous hydrogen peroxide—thereby enhancing the epoxidation process's efficiency. The absence of MgSO₄ led to a discernible decrease in both yield and enantioselectivity, as evidenced by comparative analyses presented in entries 8–10.” This amendment and the pertinent references collectively offer a comprehensive overview of the influence of MgSO₄ in our catalytic system.

These sections include references to the literature that support our findings and further elaborate on how MgSO₄ enhances the overall efficiency of the epoxidation process.

We appreciate your interest in this aspect of our study and hope that our additional references and discussion provide a clear understanding of the critical role played by MgSO₄.

Comment 2: As shown in the manuscript, the structure of 6a was established via X-ray crystallography. But only the X-ray structure of 6b was shown in Figure 3. On the other hand, there is only one chiral atom in 6a, 6a was shown as (R)-configuration instead of (S)-configuration. The authors should check the crystal data provided in SI.

Response: *Thank you for your astute observations regarding the configuration of compound 6a and the presentation of X-ray crystallography data in our manuscript. Upon reevaluation of the crystallographic data as suggested, we discovered an error in the assignment of the absolute configuration for compound 6a. We appreciate your diligence in bringing this to our attention.*

We have corrected this error and have ensured that the correct configuration, which is the (S)-configuration for 6a, is now accurately reflected in the manuscript. The revised sentence in the manuscript now reads: "The assignment of (R)-configuration to these terminal epoxides was confidently determined through a comparative analysis with 6b, the structure of which was unequivocally established via X-ray crystallography." This change corrects our previous oversight and aligns the textual data with the validated crystallographic findings.

We have also checked Figure 3 and the supplementary information to include the correct X-ray structure and configuration details for 6b to ensure clarity and accuracy in our presentation.

Thank you once again for your meticulous review and for helping us maintain the scientific accuracy and integrity of our work. We believe that these corrections have strengthened the manuscript and clarified the findings for our readers.

Comment 3: In Figure 3, the authors completed the synthesis of α -heterocyclic terminal epoxides catalyzed by (R)-4j from the vinyl-substituted N-heteroaromatic compounds without CN. In SI, the authors provide the cartesian coordinates of TS1-SR-H and Int1-SR-H without CN. If the mechanism of the reaction pathway shown in Figure 3 has been investigated through DFT calculations, it's better to provide the energy profile of the synthesis pathway shown in Figure 3.

Response: Thank you for your insightful comment and suggestion regarding the inclusion of an energy profile for the synthesis pathway shown in Figure 3. Your feedback highlights critical areas for enhancement in our manuscript and in our presentation of computational data.

In our computational investigations, we initially focused on the model compound *1aa* and its non-cyano variant to assess the impact of the cyano substituent on the reaction's energy barriers. For the compound *1aa*, a transition energy barrier of 20.4 kcal/mol from *Int1-SR* to *TS1-SR* was determined, compared to a notably higher barrier of 27.6 kcal/mol for the variant without the cyano group (from *Int1-SR-H* to *TS1-SR-H* without CN). These results are detailed in the Supplementary Information (Fig. S11) and demonstrate the significant influence of substituents on the reaction dynamics, highlighting how the presence of a cyano group can markedly lower the energy requirements of the reaction.

To contextualize these findings within the manuscript, we have revised our discussion to include: "This stabilization manifests as a substantial energy reduction of -19.6 kcal/mol (Fig. 6a), demonstrating the essential role of non-covalent interactions (NCI) in steering the reaction pathway toward high stereoselectivity. Progressing from *Int1-SR* to the final epoxide requires overcoming a free energy barrier of 20.4 kcal/mol at the transition state *TS1-SR*. In a comparative analysis, the non-cyano variant of *1aa* exhibited a notably higher barrier of 27.6 kcal/mol, illustrating the profound effect of the cyano substituent on reducing the energy requirements of the reaction. This finding underscores the significant impact of substituent nature on catalytic efficiency and stereoselectivity, effectively facilitating ring closure via *TS2-SR* and completing the epoxidation cycle."

Regarding the synthesis catalyzed by (*R*)-*4j* from the vinyl-substituted *N*-heteroaromatic compounds without CN, as illustrated in Figure 3, we have not extended our DFT calculations to include these substrates. The decision was based on our focus to elucidate the role of the cyano substituent in our primary model system, which was essential for understanding the fundamental catalytic and stereoselective interactions. Consequently, the energy profiles for reactions catalyzed by (*R*)-*4j* were not computed and thus not included in the figure.

We appreciate your suggestion and recognize the value of expanding our computational analysis to include additional substrate variants in future studies. This would provide a more comprehensive understanding of the catalytic processes across different substrate frameworks and could further validate our experimental approaches.

Thank you once again for your detailed review and constructive feedback, which are invaluable in enhancing the depth and clarity of our research presentation.

Comment 4: In Figure 5c, when $n=2$, the unit of $t \cdot [(S)-4a]^n$ is $\text{mol}^2 \cdot \text{min} \cdot \text{L}^{-2}$ instead of $\text{mol} \cdot \text{min} \cdot \text{L}^{-2}$. Please check the unit in the manuscript again.

Response: *Thank you for pointing out the discrepancy in the units used in Figure 5c of our manuscript. Upon reviewing your comment, we confirmed that the unit was indeed incorrectly stated. We appreciate your attention to detail and thank you for helping us maintain the accuracy of our scientific reporting.*

We have corrected the error and updated Figure 5c with the proper units: $\text{mol}^2 \cdot \text{min} \cdot \text{L}^{-2}$. The revised figure and corresponding text in the manuscript now accurately reflect this change, ensuring that the data presentation is correct and adheres to scientific standards.

The corrected figure and manuscript text will provide clearer and more precise information to our readers, and we are grateful for your contribution towards improving the quality of our work. Thank you again for your thorough review and for aiding in enhancing the integrity of our publication.

REVIEWERS' COMMENTS

Reviewer #1 (Remarks to the Author):

The manuscript has been revised and I agree to its publication in Nature Communications.

Reviewer #4 (Remarks to the Author):

I have thoroughly examined the initial submission of this manuscript and am pleased to report that the authors have diligently revised it, I am now very satisfied and the current version will be acceptable to me.